# Revealing Hidden Failure Modes in Chest X-ray Classification via Spectral Domain Analysis

**Samuel Halimi** ⓘD                                                    SAMUEL@AZMED.CO
**Loic Themyr**                                                         LOIC@AZMED.CO
**Arnaud Abreu**                                                        ARNAUD@AZMED.CO
*10 rue d'Uzès 75002 Paris*

## Abstract

Deep learning models for chest X-ray anomaly detection remain vulnerable to subtle distributional shifts (e.g., acquisition technique, patient-related factors, and preprocessing). Traditional error analysis often relies on semantic metadata or model embeddings, which can mask low-level signal variations that degrade performance. In this work, we propose a data-centric framework for automated failure mode discovery using spectral analysis. We project images into the frequency domain and extract a compact profile summarizing the distribution of signal energy across frequency bands. By performing unsupervised clustering on these spectral profiles, we demonstrate that model failures are not randomly distributed, but are strongly concentrated within specific spectral clusters. This method effectively isolates "blind spots", enabling the prediction of model reliability and the discovery of performance-degrading data slices without requiring ground-truth failure annotations.

**Keywords:** Chest X-ray, Failure Mode Discovery, Spectral Analysis, Model Robustness, Unsupervised Clustering.

## 1. Introduction

Deep learning has achieved remarkable success in the interpretation of chest X-rays (CXRs), where algorithms now frequently match or exceed radiologist performance in controlled settings (Katzman et al., 2024; Bettinger et al., 2024). However, in the context of clinical deployment, models often fail to generalize to data that deviates statistically from their training distribution, resulting in significant and unanticipated performance degradation (Zech et al., 2018; Yu et al., 2022). Ensuring the reliability of these systems requires the ability to anticipate and identify failure modes before they impact patient care.

Two main approaches have been explored to isolate underperforming samples from unseen data. Metadata stratification uses sensitive categorical attributes to spot biases in the system (Gichoya et al., 2023), while model-centric techniques examine the embedding of models, either to spot out-of-distribution samples (Hong et al., 2024) or to cluster coherent underperfoming slices in the datasets (Olesen et al., 2024). However, both metadata stratification and latent space analysis operate on high-level semantic abstractions and fail to capture subtle signal-level shifts and irregularities that drive performance degradation.

Therefore, in this work, we propose a *data-centric* framework that prioritizes signal characteristics over semantic features or metadata labels. Inspired by recent studies in the field of domain adaptation and generalization (Wang et al., 2020; Xu et al., 2021), we hypothesize that some systematic model failures correlate with specific profiles in the frequency domain.

From the *Fourier transform* of an image, we compute a compact representation that summarizes the distribution of signal energy across frequency bands. In that representation space, we apply an unsupervised clustering method to partition validation data into spectrally coherent subgroups. Our experiments demonstrate that these spectral clusters might act as predictors of model reliability. We show that model failures are not uniformly distributed, but are concentrated within specific spectral clusters and effectively reveal "blind spots" in the model's generalization capability. This approach allows for the identification of performance-degrading data slices rooted in image physics, enabling predictive reliability estimation without the need for ground-truth failure annotations.

## 2. Related work

### 2.1. Slice Discovery in medical image datasets

To identify systematic failures, the standard approach for medical applications consists in slicing validation data by categorical attributes (Seyyed-Kalantari et al., 2020; Larrazabal et al., 2020; Ahluwalia et al., 2023). Although essential for fairness auditing, this approach is entirely based on the availability of tabular data. It remains blind to "hidden stratification" (Oakden-Rayner et al., 2020), where performance degrades due to signal-level characteristics that are not recorded in clinical logs.

Beyond categorical attributes, Out-of-Distribution (OOD) detection uses feature space distances to spot anomalies (Lee et al., 2018; Liu et al., 2020) and is also widely explored in medical image applications (Tardy et al., 2019; Berger et al., 2021; Roy et al., 2022; Araujo et al., 2023). Although effective in identifying stark outliers, these methods struggle with subtle variations in image acquisition that degrade performance without triggering distance-based alarms (Wiles et al., 2021).

To uncover coherent high-error subsets of data, recent unsupervised methods apply clustering algorithms in the latent embedding space of models (Eyuboglu et al., 2022; d'Eon et al., 2022; Olesen et al., 2024). However, this approach faces a technical paradox: deep networks are explicitly optimized to be invariant to nuisance variables (Achille and Soatto, 2018). Consequently, the latent space often suppresses the irregularities that cause model failure, rendering these failure modes invisible to latent space clustering strategies.

### 2.2. Frequency domain analysis in deep learning

In this work, we look for failure modes in the frequency domain of the images, to remove metadata supervision and dependence on subjective model embeddings to focus only on the intrinsic irregularities of the signal. Important studies on image frequencies have been conducted in domain generalization (DG) (Huang et al., 2021a; Zhao et al., 2022) and domain adaptation (DA) (Huang et al., 2021b; Yang et al., 2022) to increase the robustness of deep learning models with respect to frequency perturbations. These methods alter either the low frequencies (Guo et al., 2018) or the high frequencies (Wang et al., 2020) of the images to train models on adversarial examples in a data augmentation scheme. Although not directly related to our approach, the success of these techniques brings evidence that some failure modes of deep learning models are explained by characteristics of the Fourier spectrum of input images.

Unlike DG and DA approaches that manipulate spectra to train robust models, we utilize spectral analysis as a *post-hoc* diagnostic tool. This allows us to cluster data based on image physics rather than semantics, exposing signal-driven failures that escape standard monitoring.

## 3. Methods

### 3.1. Radially Averaged Power Spectrum (RAPS)

We analyze signal-level variations by projecting each image into the frequency domain. For an image $x$, we compute its 2-D discrete Fourier transform $F = \mathcal{F}(x)$ and magnitude spectrum $S = |F|$. To obtain a 1-D orientation-invariant descriptor $P$, we compute the Radially Averaged Power Spectrum (RAPS) of the image following the extraction protocol by Torralba and Oliva (2003). The computation of the RAPS is illustrated in Figure 1. For a given discrete radius $r \in \mathbb{N}$, let $\Omega(r)$ be the set of frequency coordinates such as:

$$\Omega(r) = \{(u,v) \in \mathbb{Z}^2 \mid \sqrt{u^2 + v^2} = r\} \tag{1}$$

Then, for a given sampling of $N$ discrete radii $[r_1, \ldots, r_N]$, the $n$-th component $P(n)$, of the RAPS $P$, is computed as the average of the spectral magnitude $S$ over $\Omega(r_n)$:

$$P(n) = \frac{1}{|\Omega(r_n)|} \sum_{(u,v)\in\Omega(r_n)} S(u,v) \tag{2}$$

We employ the RAPS profile to compress images into compact, rotation-invariant vectors, ensuring that downstream clustering tasks focus on intrinsic statistics rather than superficial variations in object pose.

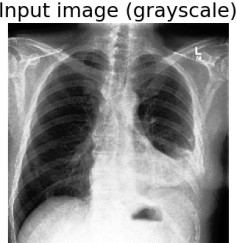 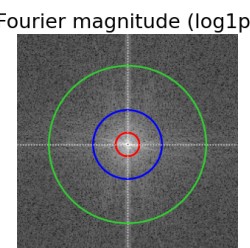 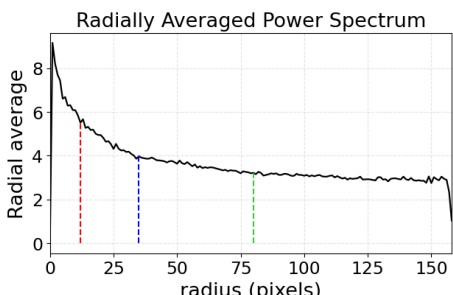

Figure 1: Computation of the RAPS of an image. We highlight the contribution of three frequency bands (red, green, blue) in the spectrum to the RAPS representation (dashed lines with corresponding colors).

To ensure comparability across images, all Fourier spectra must lie on the same frequency grid. We therefore restrict the analysis to images of similar native resolution and enforce identical spatial dimensions using minimal spatial cropping. This avoids spectral distortions: resizing modifies high-frequency content through interpolation, while padding artificially

increases low-frequency energy. A minimal uniform crop preserves the relevant spectral structure and maintains the physical interpretation of Fourier frequencies.

## 3.2. Similarity metric

For the subsequent clustering phase, we advocate the use of the correlation distance as a similarity metric between RAPS descriptors. Given the RAPS of two different unseen images, $P_i$ and $P_j$, their correlation distance is given by:

$$d_{\mathrm{corr}}(i, j) = 1 - \mathrm{corr}(P_i, P_j) \tag{3}$$

Where the correlation between $P_i$ and $P_j$ is computed across the $N$ sampled radii as:

$$\mathrm{corr}(P_i, P_j) = \frac{\sum_{n=1}^{N} \left(P_i(n) - \bar{P}_i\right)\left(P_j(n) - \bar{P}_j\right)}{\sqrt{\sum_{n=1}^{N} \left(P_i(n) - \bar{P}_i\right)^2}\sqrt{\sum_{n=1}^{N} \left(P_j(n) - \bar{P}_j\right)^2}} \tag{4}$$

Unlike the Euclidean distance, which is sensitive to absolute magnitude, the correlation distance is invariant to additive and multiplicative shifts in signal intensity. This ensures that vectors are grouped based on their intrinsic spectral morphology rather than physically irrelevant variations in global energy or sensor gain. Consequently, the clustering process effectively isolates stable structural features, such as relative sharpness (shape of the high-frequency tail), sensor or reconstruction noise patterns (boosting of high frequencies), as well as characteristic spectral slopes of specific imaging devices, while remaining robust to acquisition-level scaling and offsets.

## 3.3. K-medoids clustering

In a first exploratory phase, we use the standard K-medoids clustering to validate the relevance of RAPS descriptors for performance analysis. As it is not specifically optimized for slice discovery, it allows unbiased exploration of the relationship between the frequency domain and model failures. The possibility of varying the number of clusters is also convenient to confirm an observed tendency across partitions of different granularities.

## 3.4. Hierarchical Agglomerative Clustering (HAC)

To optimize the slice discovery process, we resort to a hierarchical clustering technique. In a bottom-up fashion, each radial profile starts as an individual cluster and we iteratively merge the closest pairs of clusters according to the average linkage rule. To partition the resulting dendrogram we set the 0.5-quantile of the distribution of pairwise distances as a threshold to stop the merging procedure. Adopting this strategy gives three key advantages. First, the partition scale naturally adapts to the inherent variability of the spectral representations. Second, it circumvents the limitation of pre-specifying an arbitrary cluster count. Finally, this flexibility effectively optimizes the isolation and discovery of specific underperforming slices.

### 3.5. Outliers

In accordance with standard agglomerative clustering protocols, outliers were defined as observations or small terminal groups that remained isolated until the final stages of the hierarchy. Specifically, we classified any cluster containing fewer than 10 samples as an outlier group, as these represented spectrally atypical profiles with linkage distances significantly exceeding the primary fusion thresholds. To maintain statistical robustness and ensure sufficient sample sizes for downstream analysis, these isolated profiles were aggregated into a single composite outlier category.

## 4. Experimental setup

### 4.1. Datasets and preprocessing

We conducted all experiments using four publicly available chest X-ray datasets commonly employed for disease classification:

**CheXpert** (Irvin et al., 2019). A large-scale dataset from Stanford containing frontal and lateral views, labeled for 14 thoracic findings using report-derived NLP. **MIMIC-CXR** (Johnson et al., 2019). A de-identified dataset from Beth Israel Deaconess Medical Center with paired radiology reports and substantial acquisition and population diversity. **Pad-Chest** (Bustos et al., 2020). A Spanish dataset with multi-view radiographs and detailed labels covering findings, diagnoses, and anatomical locations. **NIH ChestX-ray14** (Wang et al., 2017). A frontal-view dataset labeled for 14 conditions via report mining, notable for heterogeneous acquisition and label noise, making it a common robustness benchmark.

To ensure comparability of Fourier transforms and radial spectral profiles across datasets, we standardized image dimensions while preserving spectral geometry. We selected $264 \times 224$ as the target resolution, as it was the most common size across sources. Images were resized isotropically by fixing the height to 224 pixels while preserving aspect ratio. In the Fourier domain, isotropic resizing corresponds to a uniform rescaling of frequencies and avoids the geometric distortions that anisotropic resizing would introduce. This is particularly important for radial averaging, which assumes isotropic spectral structure: anisotropic scaling would induce elliptical distortions and bias the radial profile. Although interpolation during resizing may slightly attenuate high-frequency components, the procedure was applied uniformly across samples, preserving the relative spectral structure underlying our analysis. Minimal center-cropping was then performed to obtain identical final dimensions, and samples with incompatible aspect ratios were excluded to avoid excessive rescaling or truncation.

We focus on three clinically common findings: Atelectasis, Consolidation, and Pleural Effusion. We trained our models in a mono-pathology setting. For each pathology, we constructed separate training and validation splits. The training sets are exclusively based on CheXpert images. Table 1 details the datasets sizes. The test sets are built from each dataset, so MIMIC-CXR, PadChest, and NIH are unseen-domain in our evaluation. This enables the analysis of cross-dataset generalization under distributional shift.

For experiments that require large-scale validation, we derived a unified *Global* test set by pooling the test subsets from every datasets.

Table 1: The size of each mono-pathology dataset used in the study is detailed, broken down by the number of images in the training set, the seen-domain test set, and the unseen-seen test set. Note that all datasets are balanced, containing an equal number of positive and negative examples.

| Dataset | Type | Atelectasis | Consolidation | Pleural Effusion |
|---|---|---|---|---|
| **CheXpert** | Train | 39K | 39K | 39K |
| **CheXpert** | Test | 256 | 256 | 256 |
| **MIMIC** | Test | 476 | 400 | 520 |
| **NIH** | Test | 1308 | 642 | 1386 |
| **Padchest** | Test | 400 | 400 | 816 |
| **Global** | Test | 2440 | 1698 | 2978 |

### 4.2. Models, training and performance assessment

For all experiments, we trained a DenseNet-121 (Huang et al., 2018) classifier using binary cross-entropy loss and the Adam (Kingma and Ba, 2017) optimizer (learning rate $1 \times 10^{-4}$, weight decay $1 \times 10^{-3}$). A Reduce-on-Plateau scheduler (minimum learning rate $1 \times 10^{-5}$) and a batch size of 16 were used. Models were trained for up to 30 epochs with early stopping based on validation performance. All training was performed on an NVIDIA GeForce RTX 3080 GPU.

We report the Area Under the Receiver Operating Characteristic (AUROC) to evaluate the discriminative performance of the models. This metric was chosen for its invariance to decision thresholds and its robustness to the fluctuating prevalence in unsupervised clustering assignments.

## 5. Experiments and results

### 5.1. Relevance of RAPS descriptors for performance analysis

Given a K-medoids partition of the RAPS descriptors (Section 3.3), we use the standard deviation of the AUROC across the $K$ clusters to measure how effectively the partition stratifies performance. To isolate the effect of spectral coherence from artifacts of cluster size or class distribution, we implement two stochastic baselines: a *fully random* (FR) assignment and a *pseudo-random* (PR) control that preserves the specific cluster size and label statistics of the K-medoids solution. To study the impact of granularity, we repeat the above experiment for values of $K$ ranging from 2 to 10. We then average the results over 50 seeds to reduce variability from K-medoids initialization and baseline stochasticity.

To demonstrate the consistency of our findings across varying model architectures, we replicated the experiments using both ResNet (He et al., 2015) and EfficientNet (Tan and Le, 2020) architectures; these results are detailed in Appendix B.

As shown in Figure 2, the dispersion of AUROC in the K-medoids partitions of RAPS profiles (blue) remains consistently above those of both random baselines, sometimes reaching nearly twice as high. Importantly, the pseudo-random baseline (red) preserves the exact

cluster size distribution and class proportions of the K-medoids solution, while discarding any organization based on spectral similarity. This control ensures that the observed variability cannot be attributed merely to differences in cluster size or label imbalance.

Moreover, all results are averaged over 50 random seeds to further reduce stochastic effects arising from K-medoids initialization and from random cluster composition. The persistent and substantially larger gap between the blue and red curves indicates that the partitions induced by the correlation distance between RAPS profiles capture genuine heterogeneity in model behavior.

The consistency of this effect across all pathologies (see Appendix A), across different classifier architectures (ResNet, EfficientNet, and DenseNet; Appendix B), and for all values of $K$ reinforces our confidence that spectral representations derived from Fourier analysis explain a meaningful portion of the observed performance variability.

Because the correlation distance emphasizes shape differences, this gap suggests that differences in spectral composition—such as the relative distribution of low versus high frequencies, decay rates, or local spectral slopes—are systematically associated with differences in model performance.

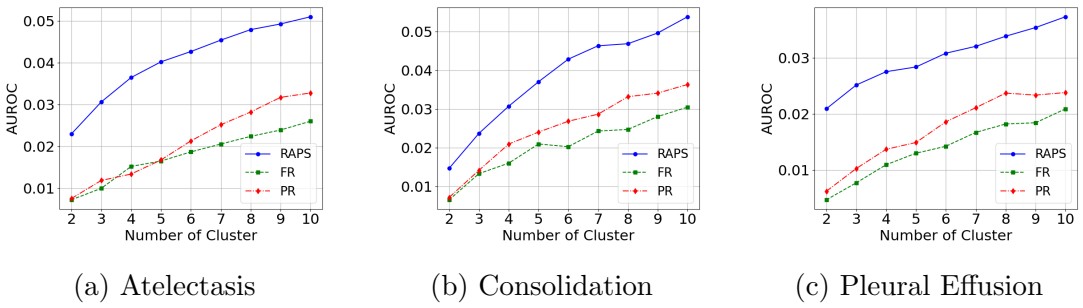

(a) Atelectasis        (b) Consolidation        (c) Pleural Effusion

Figure 2: AUROC variability across different number of clusters ($K$) made by the k-medoids method with the correlation metric vs. clusters made by fully and pseudo random methods. Results are averaged over 50 seeds. Clustering is performed on the Global aggregated test set.

## 5.2. Influence of the similarity metric on performance stratification

To assess how the distance metric influences the K-medoids performance stratification, we compare three metrics on the radial profiles: Euclidean, Spearman, and correlation. The evaluation follows the same protocol as in Section 5.1, with AUROC variability compared to the FR and PR baselines and averaged over multiple seeds.

Across all values of $K$, the correlation distance produces the largest separation between the clustering curve and both baselines (Figure 3). The Spearman distance induces greater variability than the Euclidean distance, though still below the correlation distance.

By contrast, the $L_2$ distance shows only limited gains over the baselines, suggesting that amplitude-based similarities contribute little to the stratification of performance compared to differences in the *shape* of the spectral profiles. This ordering among metrics indicates

that the performance variability is more closely related to the *relative distribution* of spectral energy than to its absolute magnitude. Metrics that are invariant to global scaling and affine transformations, such as the correlation distance and, to a lesser extent, the Spearman distance, capture spectral characteristics that are more informative for distinguishing model behavior. This supports the choice of the correlation distance in our clustering framework and is consistent with Section 5.1, where it produces the largest performance disparities across clusters.

These findings suggest that DG and DA strategies based on the frequency domain should not focus exclusively on matching absolute spectral energy distributions, as is often done in methods that impose target spectra or mix frequency components directly. Instead, our results indicate that the *relative* structure of the radial spectrum, such as the balance between low and high frequencies and the rate at which spectral energy decays, plays a more consequential role. Methods that explicitly account for these relative spectral characteristics may therefore offer a more effective direction for adaptation and robustness.

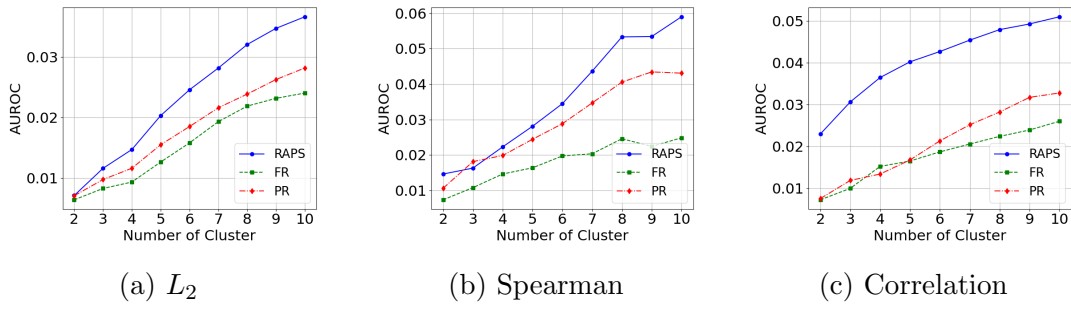

(a) $L_2$  (b) Spearman  (c) Correlation

Figure 3: Ablation study evaluating the influence of the distance metric used in k-medoids clustering on AUROC variation for the Atelectasis pathology. We compare three metrics: Euclidean ($L_2$) distance, Spearman distance, and the correlation distance employed in our proposed method.

### 5.3. Slice discovery and outlier detection

To confirm the robustness of our data-centric framework, we apply HAC in the space of RAPS descriptors to isolate under-performing data slices on unseen domain data. After clustering, we assess the AUROC on each cluster to pinpoint specific spectral regions where model performance deviates from the norm.

As depicted in Figure 4, the AUROC across HAC clusters exhibits distinct and variable performance levels for all pathologies and external validation centers. This finding mirrors the performance stratification observed with the K-medoids clustering approach. This convergence confirms that the spectral differences encoded through the correlation distance reliably capture meaningful variations in model performance, regardless of the underlying clustering algorithm employed.

Importantly, we demonstrate that across almost all considered datasets and pathologies, the proposed method systematically isolates a cluster exhibiting significantly degraded

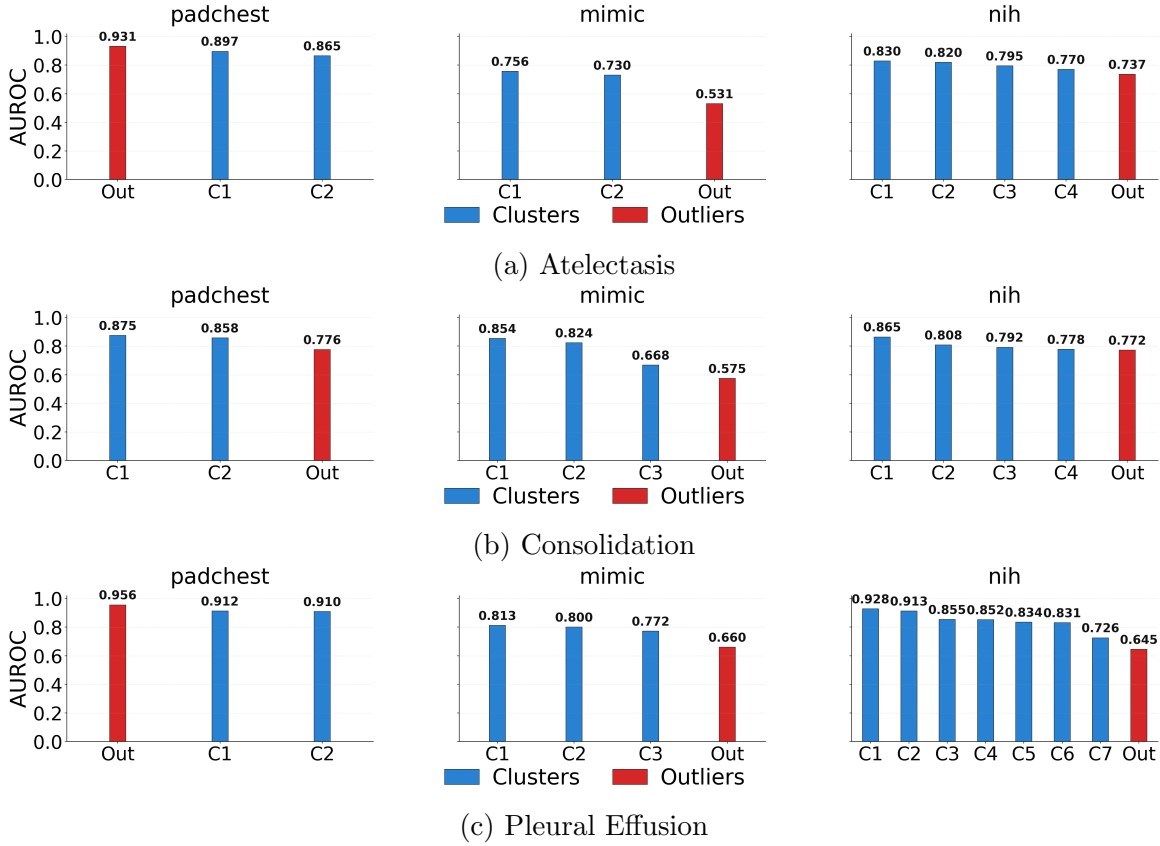

(a) Atelectasis

(b) Consolidation

(c) Pleural Effusion

Figure 4: Comparative analysis of model performance (AUROC) across data clusters (blue) defined by HAC. The figure highlights the distinct model reliability on the identified clusters versus the significantly different performance (red) observed on the HAC-isolated outlier set. Results on the unseen-domain test datasets.

AUROC (bars on the far right of the different plots). This consistency underscores the efficacy of applying HAC to RAPS descriptors for slice discovery, as it reliably uncovers latent subgroups where model performance is compromised.

Additionally, a particularly characteristic and instructive pattern emerged from the analysis of the outlier super-group (red), which was formally defined by merging all HAC clusters containing fewer than 10 samples. The AUROC for this super-group consistently exhibited a marked separation from the performance mean of the main clusters. Specifically, the performance was frequently significantly lower; as observed, for example, in Atelectasis and Pleural Effusion on the MIMIC and NIH datasets, and in Consolidation on Padchest and MIMIC; indicating substantial failure concentration. Conversely, in select cases, the AUROC was noticeably higher (e.g., Atelectasis and Pleural Effusion on Padchest), but critically, it was never comparable to the average cluster performance.

As suggested by the analyses in Appendix C and D, these variations in performance are not readily attributable to observable visual or demographic differences alone. Therefore, this consistent performance separation provides strong quantitative evidence: slices possessing atypical spectral signatures reliably translate into performance-level outliers within the model.

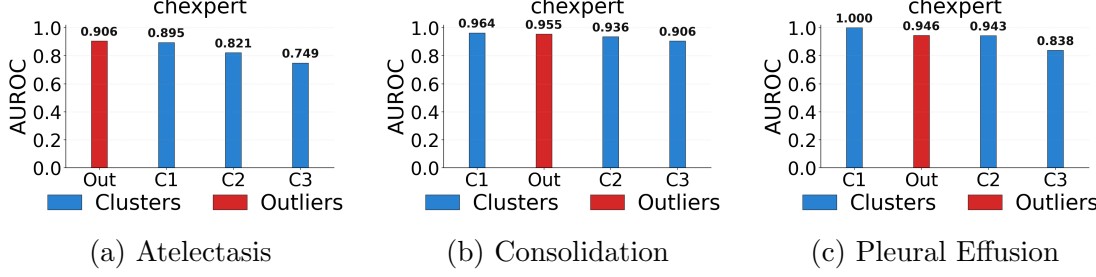

Figure 5: Comparative analysis of model performance (AUROC) across data clusters (blue) defined by HAC. The figure highlights the distinct model reliability on the identified clusters versus the significantly different performance (red) observed on the HAC-isolated outlier set. Results on seen-domain test datasets: CheXpert.

We also evaluated the model on seen-domain images with the CheXpert test dataset. As shown in Figure 5, the performance on the outlier super-group is not markedly different from that of the other clusters. This outcome is expected: because the model was trained exclusively on this domain, it learned to handle the full range of RAPS variations present in the training distribution. Nevertheless, we still observe clusters with distinct performance levels, indicating that HAC can partition the domain into meaningful RAPS subsets on which the model performs comparatively better or worse.

## 5.4. Comparison with OOD detection baselines

Beyond internal performance stratification, we further evaluate whether the proposed spectral framework can effectively identify performance-degrading samples in an out-of-distribution (OOD) setting. To this end, we compare RAPS-based outlier detection against established

OOD baselines: (Lee et al., 2018) and (Tardy et al., 2019). Table 2 presents the AUROC difference ($\Delta$) relative to the global test set for each experiment. Our results show that HAC of RAPS more consistently isolates OOD clusters with performance levels significantly lower than the baseline. This trend is further evidenced by the mean absolute difference, where HAC of RAPS achieves 0.103, compared to 0.077 and 0.053 for the baselines. Importantly, while both SOTA methods are supervised, HAC of RAPS is fully unsupervised, making it independent to the annotation or borderline cases.

Table 2: Performance Baselines (Test) and Method Deltas (OOD - Test). The $\Delta$ represents the difference between the AUROC on detected outliers and the global test set baseline.

| | Consolidation | | | Atelectasis | | | Pleural Effusion | | | AVG |
|---|---|---|---|---|---|---|---|---|---|---|
| Method | Pad. | MIM. | NIH | Pad. | MIM. | NIH | Pad. | MIM. | NIH | $|\Delta|$ |
| Test (Baseline) | 0.848 | 0.765 | 0.793 | 0.888 | 0.739 | 0.778 | 0.920 | 0.787 | 0.835 | |
| HAC of RAPS ($\Delta$) | **-.072** | **-.190** | -.021 | +.043 | **-.209** | -.041 | +.036 | **-.127** | **-.190** | **0.103** |
| (Lee et al., 2018) ($\Delta$) | +.023 | +.076 | +.029 | **-.124** | -.139 | **+.110** | -.063 | -.052 | -.081 | 0.077 |
| (Tardy et al., 2019) ($\Delta$) | -.053 | +.033 | **+.039** | -.087 | -.032 | +.049 | **-.066** | -.099 | -.071 | 0.053 |

## 6. Conclusion

In this work, we introduced a data-centric framework that uses Fourier-domain analysis to examine performance variability in chest X-ray classifiers. Representing each image through its Radially Averaged Power Spectrum (RAPS) yielded model-agnostic descriptors capturing acquisition-level signal properties. Across experiments, K-medoids and hierarchical clustering consistently revealed meaningful differences in model behavior across groups defined by their spectral profiles, with hierarchical clustering further isolating small sets of images whose atypical frequency patterns were matched by equally atypical performance across evaluation domains.

Ablations over similarity metrics clarified which components of the spectral representation matter most. Correlation distance produced the strongest separation between clusters, indicating that performance differences relate primarily to the relative distribution of spectral energy, its shape and rate of decay, rather than to absolute magnitude. This identifies the spectral characteristics most relevant to model behavior and motivates approaches that explicitly account for such relative frequency patterns.

Together, these findings point to limitations of metadata- or embedding-based slice discovery. Signal-level irregularities rooted in acquisition physics, preprocessing, or frequency composition form predictive patterns that standard monitoring pipelines overlook. Spectral analysis therefore provides a principled and interpretable means of exposing these hidden strata and improving performance auditing. Beyond error analysis, our results suggest that robustness and adaptation strategies may benefit from targeting spectral structure directly, particularly the balance between low and high frequencies, offering a more physically grounded avenue for improving generalization in medical imaging.

## Acknowledgments

The authors thank Alexandre Attia, Julien Vidal and Elie Zerbib for supporting this work at AZmed.

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

# Appendix A. Ablation extension

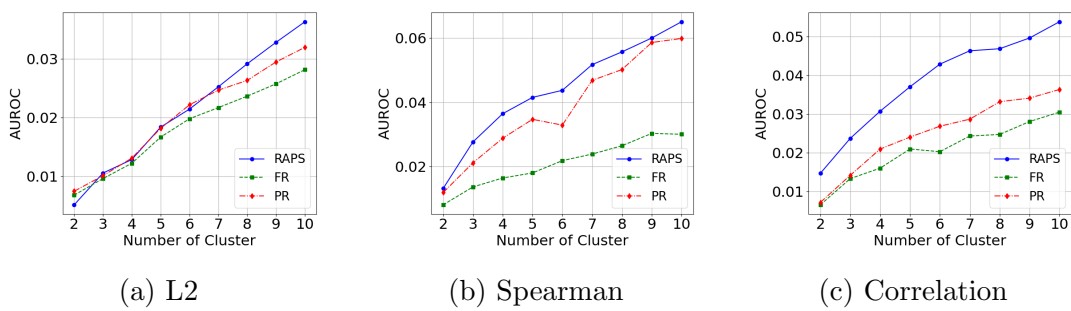

(a) L2         (b) Spearman        (c) Correlation

Figure 6: Ablation study evaluating the influence of the distance metric used in k-medoids clustering on AUROC variation for the Consolidation pathology. We compare three metrics: Euclidean ($L_2$) distance, Spearman distance, and the correlation distance employed in our proposed method.

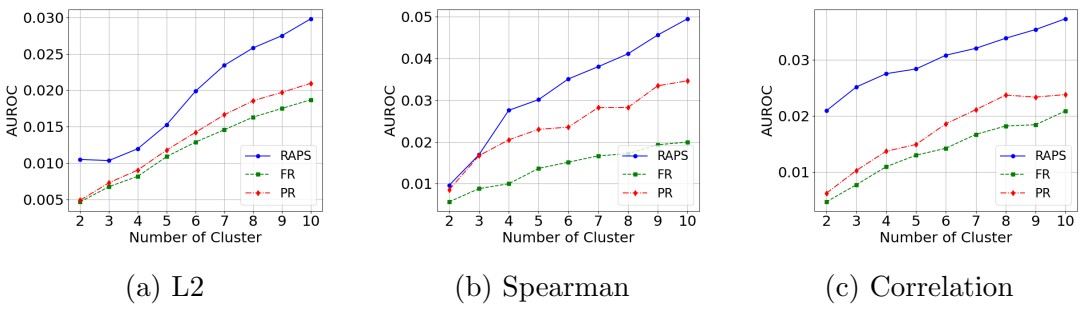

(a) L2         (b) Spearman        (c) Correlation

Figure 7: Ablation study evaluating the influence of the distance metric used in k-medoids clustering on AUROC variation for the Pleural Effusion pathology. We compare three metrics: Euclidean ($L_2$) distance, Spearman distance, and the correlation distance employed in our proposed method.

## Appendix B. Results with other models

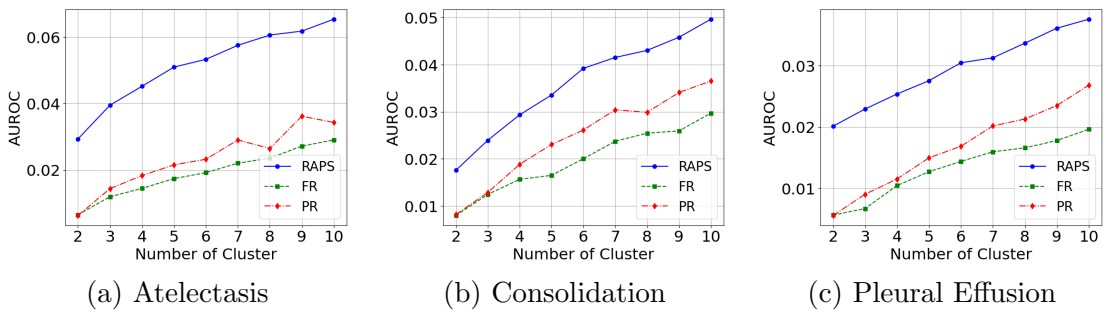

(a) Atelectasis      (b) Consolidation      (c) Pleural Effusion

Figure 8: RAPS evaluated on Resnet50 model (He et al., 2015). AUROC variability across different number of clusters $(K)$ made by the k-medoids method with the correlation metric vs. clusters made by fully and pseudo random methods. Results are averaged over 50 seeds.

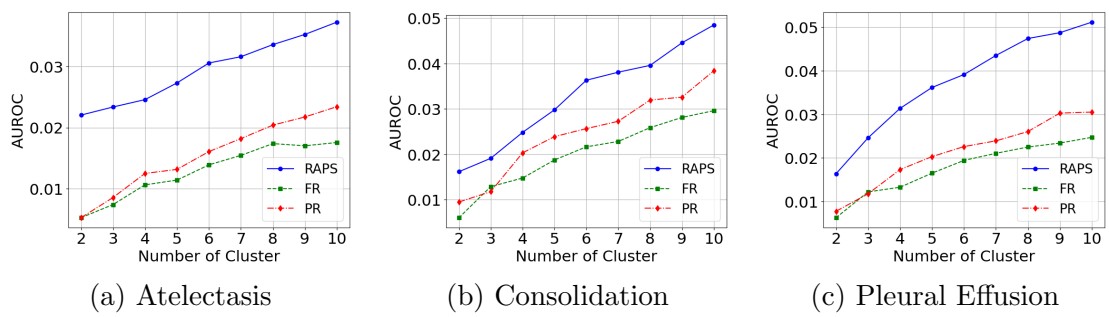

(a) Atelectasis      (b) Consolidation      (c) Pleural Effusion

Figure 9: RAPS evaluated on EfficientNet model (Tan and Le, 2020). AUROC variability across different number of clusters $(K)$ made by the k-medoids method with the correlation metric vs. clusters made by fully and pseudo random methods. Results are averaged over 50 seeds.

## Appendix C. Comparison with semantic metadata

For Pleural Effusion cases from the NIH center, we compare the AUROC of the worst slice isolated by our method with traditional slicing by available metadata attributes, i.e. age, gender, and view. As demonstrated in Table 3, metadata slicing sometimes fails to identify performance-degrading subsets, resulting in only minor AUROC variations. In contrast, our data-centric framework successfully isolates a highly vulnerable data slice that exhibits a substantial and critical performance drop. This result definitively establishes our method as an effective complementary solution for automated failure mode discovery.

Table 3: Comparison of model performance achieved by slicing the data using conventional semantic metadata against the proposed HAC of RAPS method. Results are presented for the Pleural Effusion pathology within the NIH dataset. (P: Posterior; A: Anterior)

| Category | Subgroup | AUROC |
|---|---|---|
| Age | < 20 | 0.863 |
| | 20–60 | 0.832 |
| | > 60 | 0.839 |
| Gender | Female | 0.856 |
| | Male | 0.824 |
| View | AP | 0.827 |
| | PA | 0.845 |
| **Proposed** | **Our Slice** | **0.645** |

# Appendix D. Qualitative analysis

**outliers :**

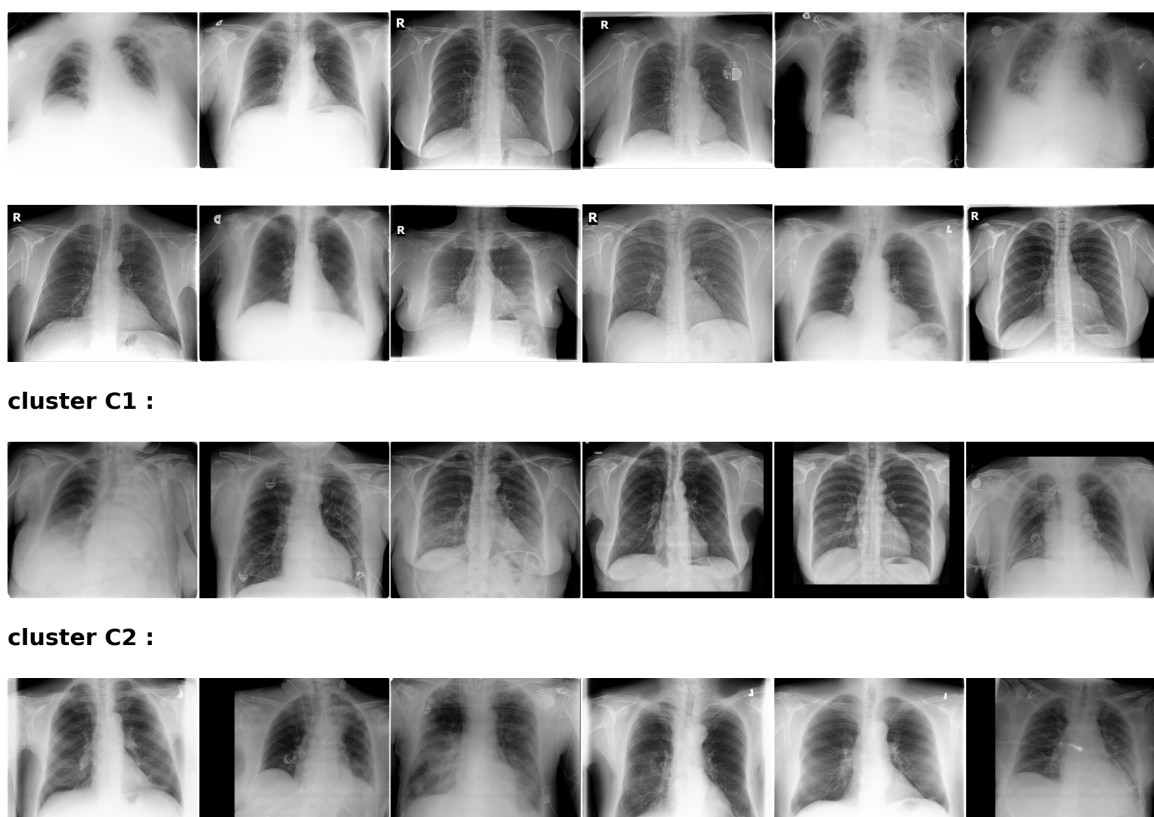

**cluster C1 :**

**cluster C2 :**

Figure 10: Qualitative results on Atelectasis for Padchest.

**outliers :**

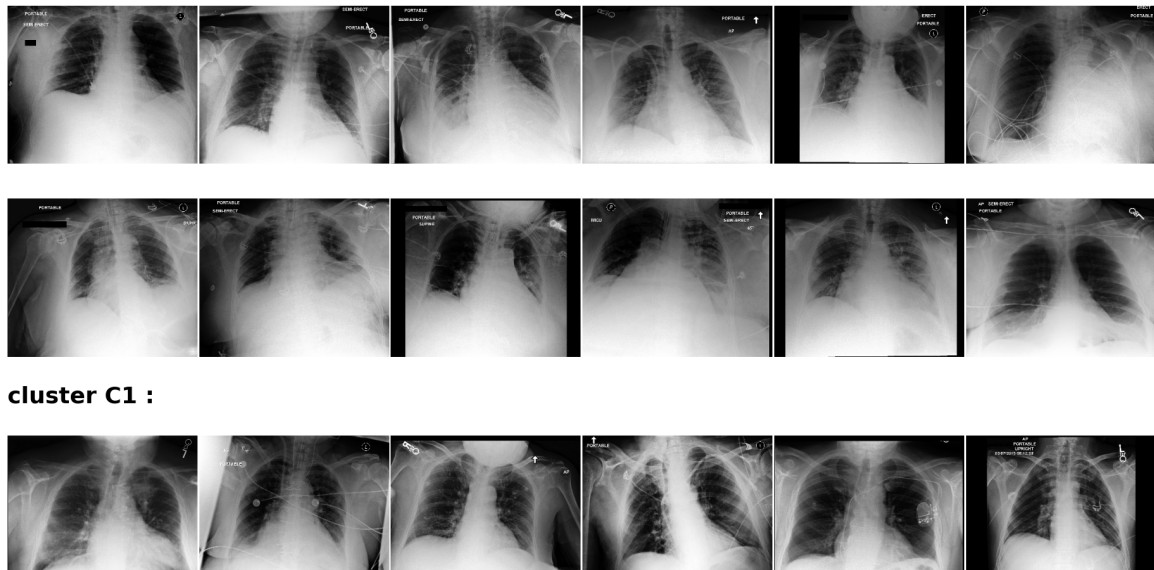

**cluster C1 :**

**cluster C2 :**

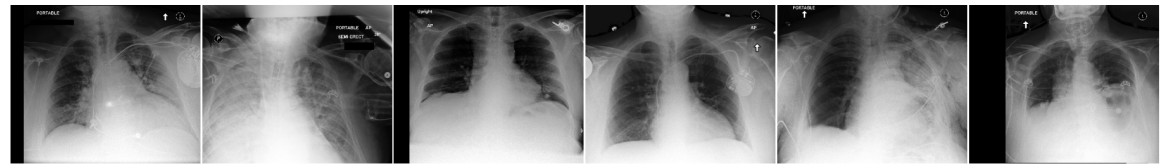

Figure 11: Qualitative results on Atelectasis for MIMIC.

**outliers :**

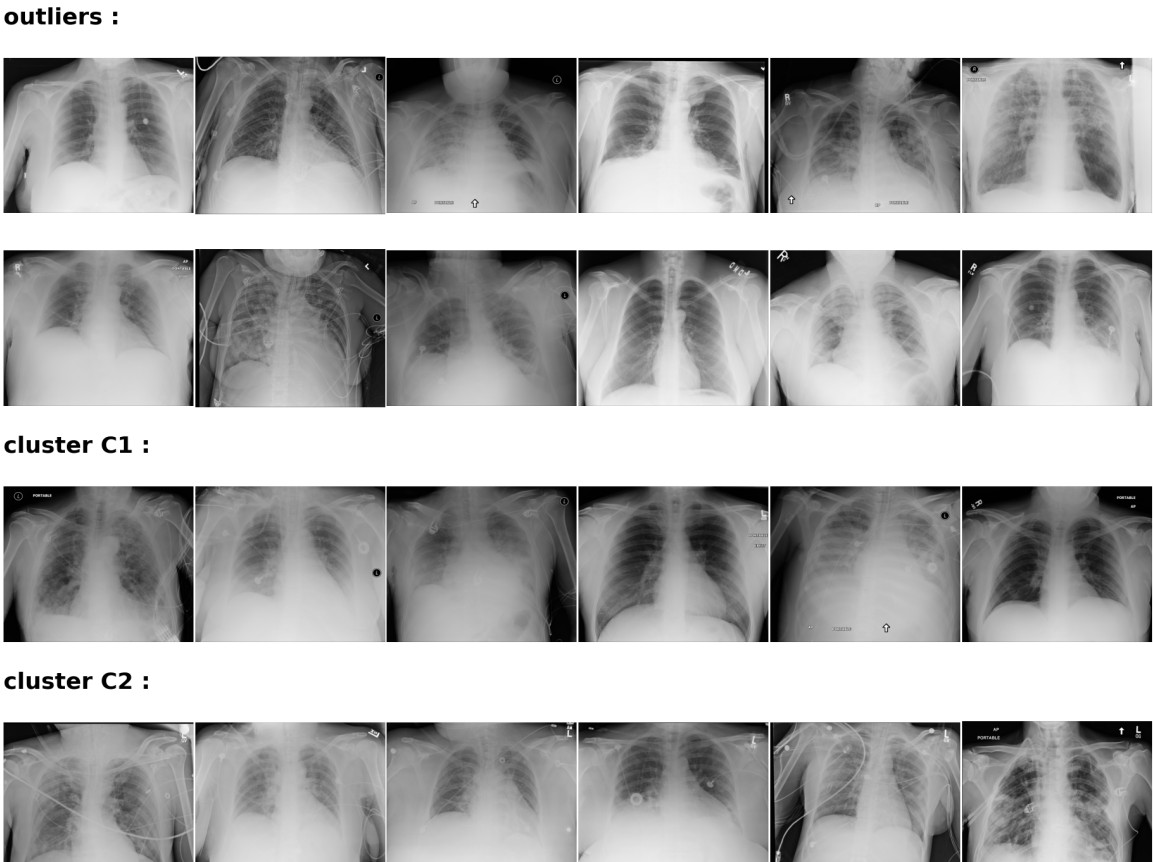

**cluster C1 :**

**cluster C2 :**

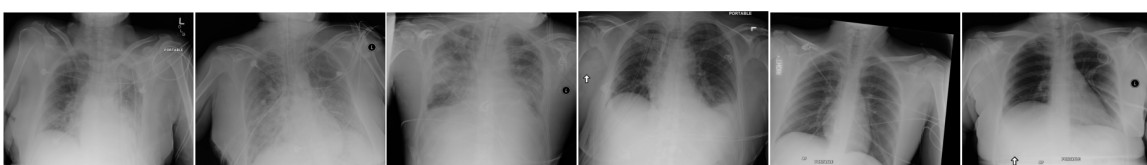

**cluster C3 :**

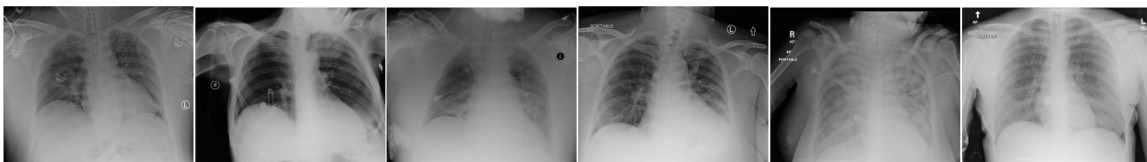

**cluster C4 :**

Figure 12: Qualitative results on Atelectasis for NIH.

**outliers :**

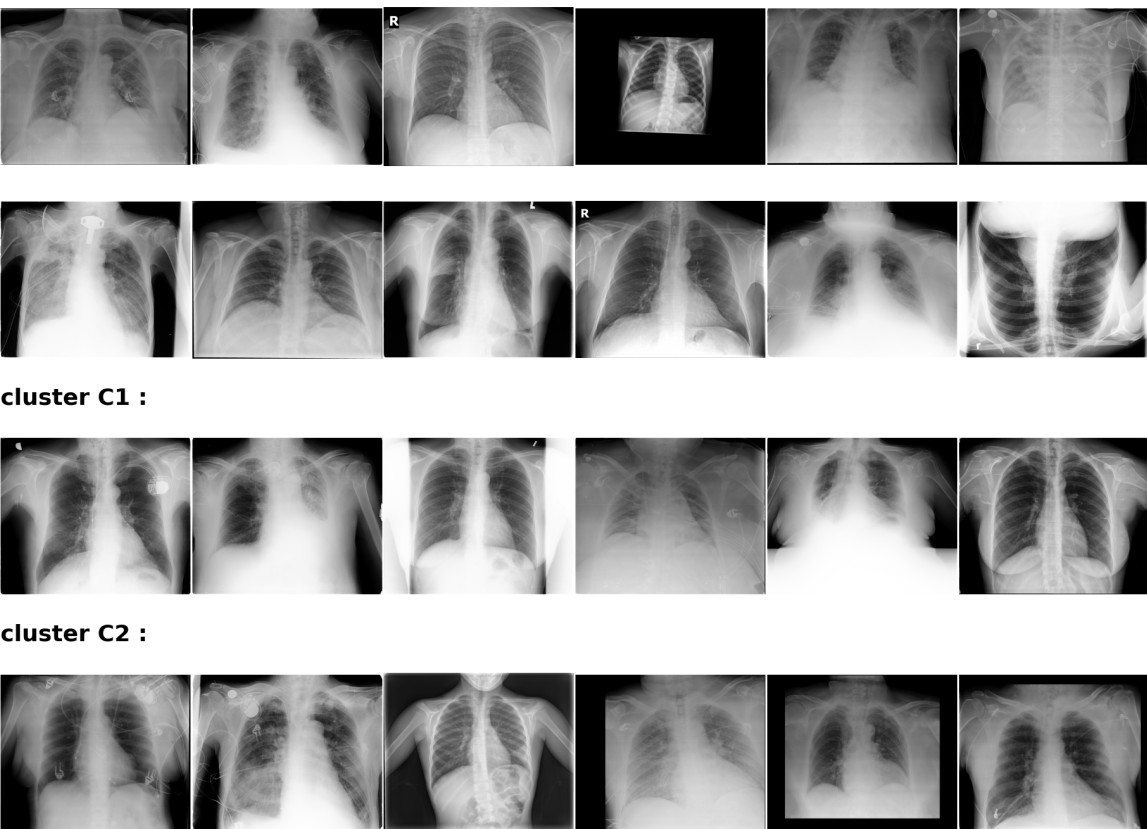

**cluster C1 :**

**cluster C2 :**

Figure 13: Qualitative results on Consolidation for Padchest.

**outliers :**

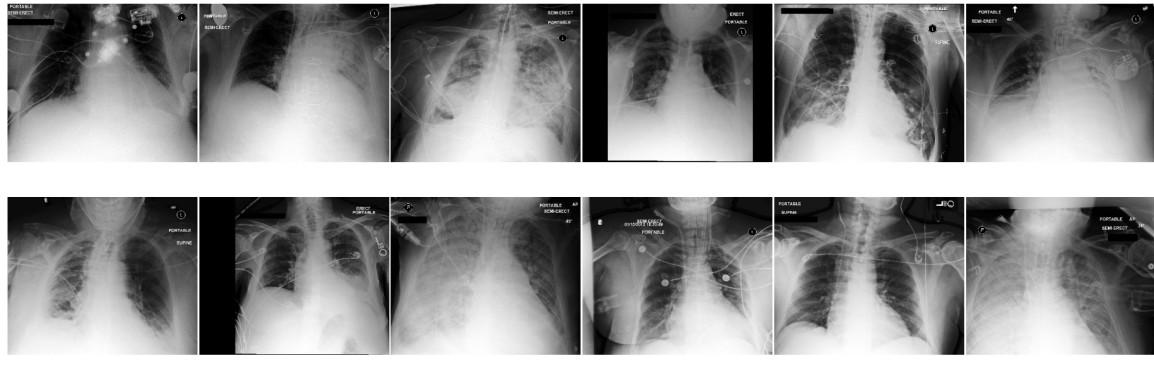

**cluster C1 :**

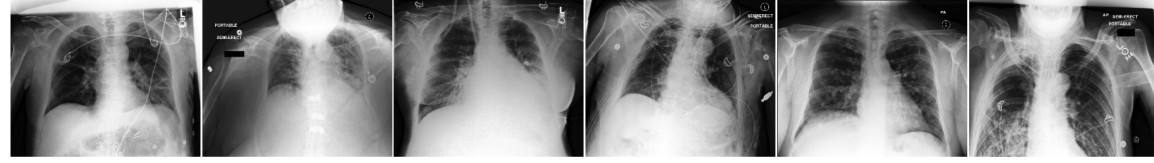

**cluster C2 :**

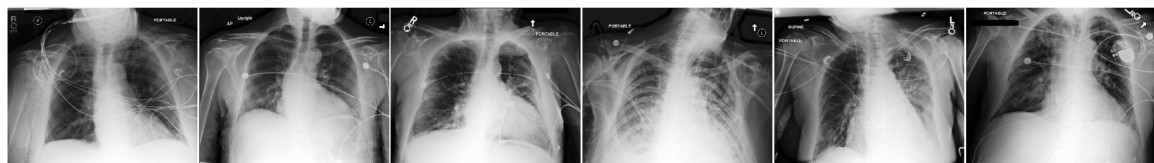

**cluster C3 :**

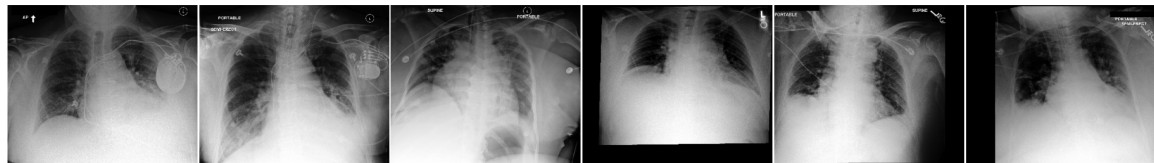

Figure 14: Qualitative results on Consolidation for MIMIC.

**outliers :**

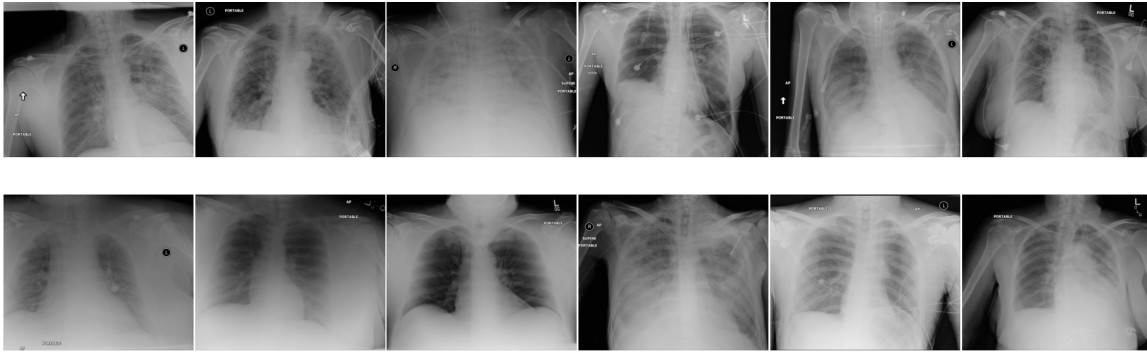

**cluster C1 :**

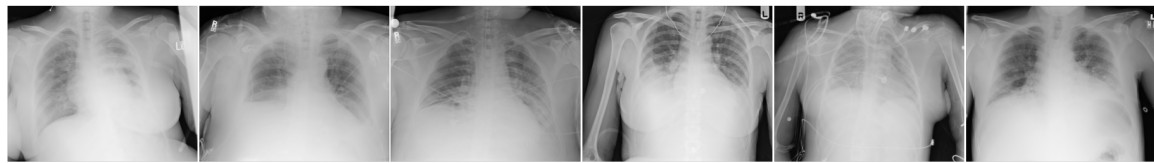

**cluster C2 :**

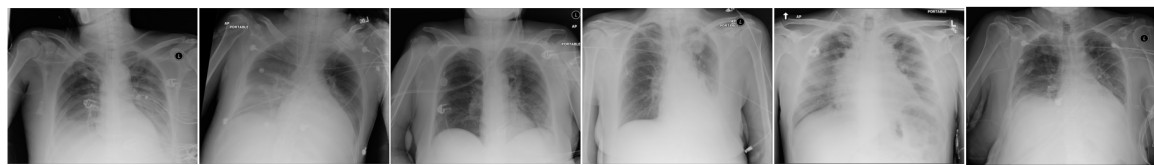

**cluster C3 :**

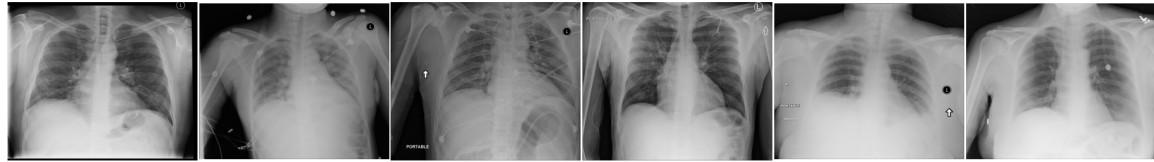

**cluster C4 :**

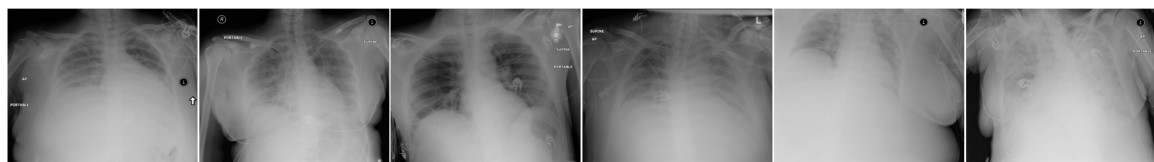

Figure 15: Qualitative results on Consolidation for NIH.

**outliers :**

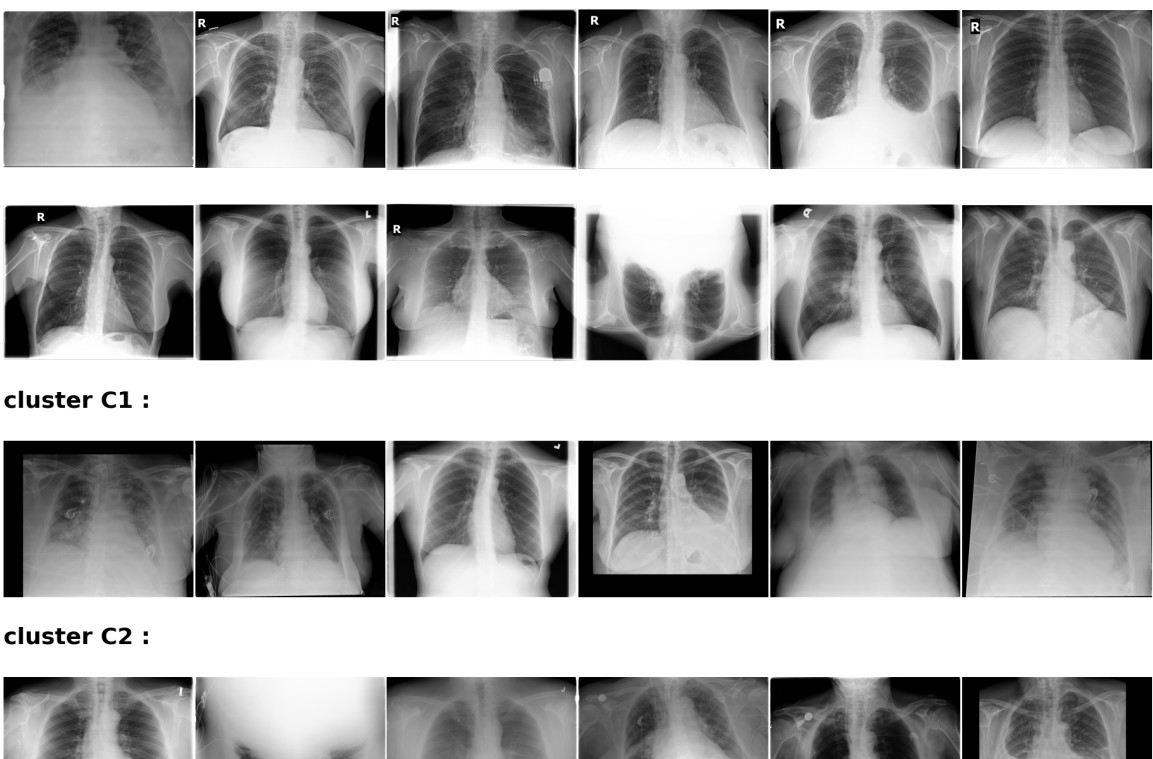

**cluster C1 :**

**cluster C2 :**

Figure 16: Qualitative results on Pleural Effusion for Padchest.

**outliers :**

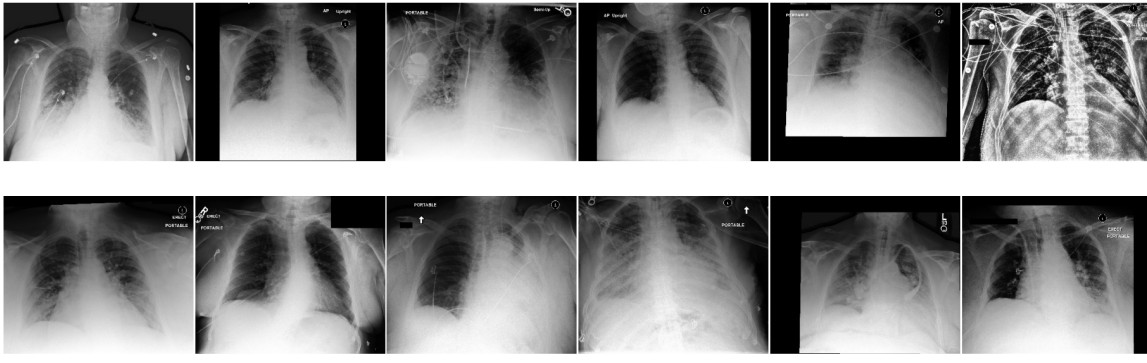

**cluster C1 :**

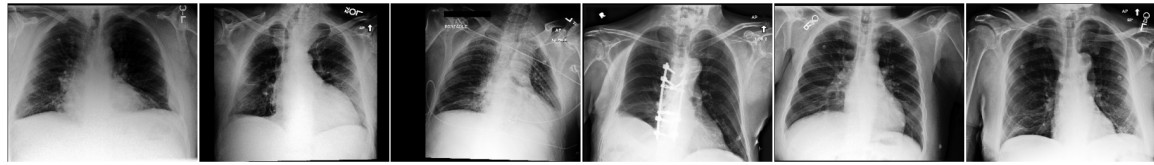

**cluster C2 :**

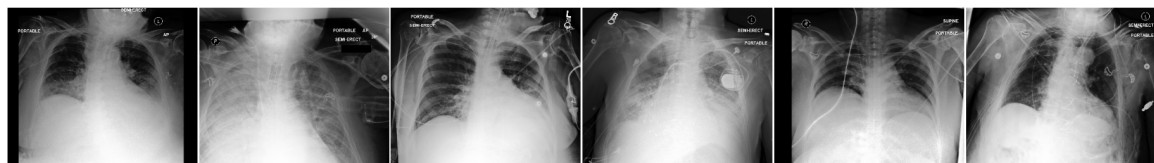

**cluster C3 :**

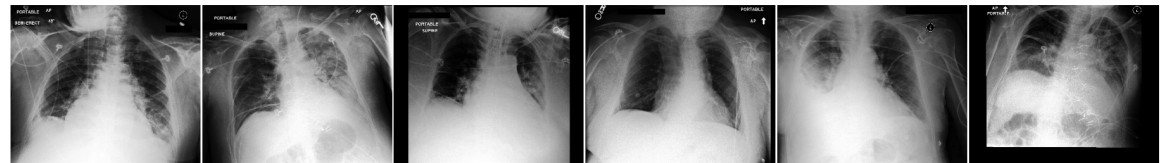

Figure 17: Qualitative results on Pleural Effusion for MIMIC.

**outliers :**

**cluster C1 :**

**cluster C2 :**

**cluster C3 :**

**cluster C4 :**

**cluster C5 :**

**cluster C6 :**

**cluster C7 :**

Figure 18: Qualitative results on Pleural Effusion for NIH.

