# OpenReview forum: "Revealing Hidden Failure Modes in Chest X-ray Classification via Spectral Domain Analysis"
_MIDL.io/2026/Conference — MIDL 2026 Poster_

### Official Review · Reviewer_7oqB · 2025-12-29

**Confidence:** 3
**Preliminary Rating:** 4
**Final Rating:** 4

**Summary:**

In this paper, the authors propose a data-centric framework for discovering failure modes in chest X-ray classification models by analyzing images in the frequency domain. The authors use the Radially Averaged Power Spectrum (RAPS) and create descriptors that summarize the distribution of signal energy across frequency bands. In this spatial representation of the descriptors, unsupervised clustering with correlation distance is applied, showing that model failures are concentrated within specific spectral clusters. The experiments use a DenseNet-121 trained on CheXpert and evaluated on multiple unseen datasets (MIMIC-CXR, NIH, PadChest) for three pathologies (atelectasis, consolidation, pleural effusion). The results reveal that spectral clustering successfully isolates underperforming data slices compared to classical metadata-based techniques. In this way, the proposed method enables automatic failure mode discovery.

**Strengths:**

- The use of Fourier analysis for failure mode discovery in medical imaging is an innovative idea and is clearly justified. The problem is well presented, especially the observation that deep networks suppress irregularities in latent space because they are optimized for invariance to nuisance variables.

- The experiments are well designed and include four datasets and three pathologies. The use of two types of baselines (fully random and pseudo-random preserving cluster statistics) consolidates the final results. Furthermore, ablation studies show the correct choice of correlation distance over Euclidean and Spearman metrics.

- In general, the selection and use of metrics throughout the paper are well justified and clearly explained.

- The paper is well written and includes diagrams that facilitate a better understanding. Furthermore, the methodology is well justified and well articulated, and the mathematical formulation is provided.

- The literature is concise and allows the paper to be situated in the context of slice discovery and frequency domain analysis.

**Weaknesses:**

- In this paper, the authors only compare the proposed methodology with random baselines and metadata slicing. However, a comparison with state-of-the-art methods is needed, such as works based on latent embedding spaces (Olesen et al. 2024).

- The experiments were conducted using DenseNet-121; however, the selection of this architecture over others should be better justified. Furthermore, other architectures should also be evaluated, as they may exhibit different behaviors, in order to more effectively generalize the results across multiple models.

- In Table 2, the authors only show one comparison with one pathology and using a single dataset, it is necessary to show multiple cases where spectral clustering finds failures missed by metadata.

**Detailed Comments:**

- It would be useful to add a comparison of the proposed methodology with state-of-the-art methods.

- I recommend including at least one additional architecture other than DenseNet-121 (e.g., ResNet) to demonstrate that the proposed methodology is architecture-agnostic.

- It would be great to add images showing RAPS profiles for the performance of the different clusters.

- I recommend expanding Table 2 to include more datasets and pathologies.

**Justification Of Final Rating:**

The paper is definitely interesting and introduces a novel methodology based on Fourier analysis for failure mode discovery. However, it has some drawbacks that have been partially clarified. Thanks to the authors for their responses; the extension of the paper and additional experiments strengthen it. Finally, I believe I will maintain my original score.

**Justification Of The Preliminary Rating:**

The paper presents an interesting and novel methodology based on Fourier analysis for failure mode discovery with solid experimental validation across multiple datasets. However, it lacks comparisons with state-of-the-art work, does not sufficiently demonstrate that it is architecture-agnostic, among other drawbacks. Overall, the main contribution of the paper is valuable, but it needs some improvement.

**Questions To Address In The Rebuttal:**

- How does the proposed method perform with different architectures (e.g., ResNet or EfficientNet)?

- What specific spectral characteristics differentiate failing clusters?

---

> ### Author Response · Authors · 2026-01-23
>
> **How does the proposed method perform with different architectures (e.g., ResNet or EfficientNet)?**
>
> To provide a comprehensive assessment of our method’s relative performance, we have significantly expanded our results section with two major additions:
>
> - Benchmarking against State-of-the-Art (SOTA): We have included a comparative study with established outlier detection methods (e.g., Tardy et al. MICCAI 2019 and Lee et al. NeurIPS 2018). The results are summarized in a new table comparing the performance gap (drop in AUROC) between the global test set and the outliers detected by each method. Our RAPS-based approach demonstrates a larger and more consistent performance drop across most cases, indicating its superior ability to identify true model failure modes compared to existing techniques. Notably, unlike the two SOTA methods presented, our method is fully unsupervised, which further enhances its utility and ease of deployment.
>
> - Architectural Robustness: To demonstrate that our findings are not architecture-specific, we repeated the experiments regarding performance separation (Section 5.1, Figure 2) using two additional backbones: ResNet and EfficientNet. These experiments yielded identical conclusions, confirming the robustness of the RAPS descriptor. The corresponding figures have been added to the Appendix of the revised manuscript.
>
> **What specific spectral characteristics differentiate failing clusters?**
>
> Our clustering procedure is based on correlation distance between radial spectral profiles. As such, cluster formation depends on the overall *shape* of the spectrum rather than on isolated frequency bands. Correlation emphasizes relative differences in spectral energy distribution, meaning that clusters are defined by global interactions between low-, mid-, and high-frequency components rather than by a single dominant frequency range.
>
> We attempted band-specific or pointwise analyses of the spectrum, but these did not reveal consistent discriminative peaks across datasets. This suggests that performance heterogeneity is driven by integrated spectral patterns rather than by isolated frequency components. A more detailed investigation of these interactions would be needed to fully characterize the underlying mechanisms.
>
> Regarding the suggestion to include profile visualizations, this lack of distinct frequency peaks is a reason why we could not provide a clear visual representation for the outliers. Because the clustering is driven by relative variations across a large number of samples rather than specific peaks in certain frequency bands, the average profiles do not yield visually discriminative features that can be easily plotted or interpreted at a glance.
>
> - Architectural Robustness: To demonstrate that our findings are not architecture-specific, we repeated the experiments regarding performance separation (Section 5.1, Figure 2) using two additional backbones: ResNet and EfficientNet. These experiments yielded identical conclusions, confirming the robustness of the RAPS descriptor. The corresponding figures have been added to the Appendix of the revised manuscript.
>
> - Benchmarking against State-of-the-Art (SOTA): We have included a comparative study with established outlier detection methods (e.g., Tardy et al. MICCAI 2019 and Lee et al. NeurIPS 2018). The results are summarized in a new table comparing the performance gap (drop in AUROC) between the global test set and the outliers detected by each method. Our RAPS-based approach demonstrates a larger and more consistent performance drop across most cases, indicating its superior ability to identify true model failure modes compared to existing techniques.
>
> **In Table 2, the authors only show one comparison with one pathology and using a single dataset, it is necessary to show multiple cases where spectral clustering finds failures missed by metadata.**
>
> You are right. In the revised manuscript, we have therefore moved this analysis to Appendix C and expanded it with a qualitative examination of the images in Appendix D, covering multiple clusters and datasets. Across these cases, we do not observe clear or systematic visual differences that would readily explain the performance gaps. This further supports the view that our approach provides complementary insights beyond what can be captured through metadata-based slicing alone.

---

> > ### Comment · Reviewer_7oqB · 2026-01-29
> >
> > Many thanks to the authors for their responses, which have clarified several points.

---

### Official Review · Reviewer_s6ms · 2026-01-08

**Confidence:** 5
**Preliminary Rating:** 1
**Final Rating:** 3

**Summary:**

This work introduces a spectral domain approach for identifying outliers/failure modes in chest X-ray classification models. Specifically, the authors propose representing each image using a radially averaged power spectrum, which captures its frequency domain characteristics in a compact form. These spectral representations are then compared using the correlation coefficient as a similarity measure for effective clustering of images. The resulting clusters are further analyzed to detect outliers and systematically characterize distinct failure modes of the classifier.

**Strengths:**

1) The proposed method is presented in a clear and well-structured manner, conveying the key ideas succinctly while providing sufficient detail to understand the motivation, formulation, and implementation.

**Weaknesses:**

1) The experimental validation of the proposed idea is limited and lacks sufficient depth to convincingly demonstrate the effectiveness and robustness of the approach.

2) The paper does not include a comparison with established state-of-the-art methods. Instead, the evaluation is restricted to a few method-specific baseline models, which limits the assessment of the proposed approach’s relative performance.

3) The results obtained using the proposed method are not fully consistent. For example, in some cases, the AUROC for the identified outlier cases is higher than that of other clusters, which raises questions about the reliability of the RAPS descriptors and the clustering process for the outlier detection.

4) A qualitative visual assessment of the out-of-distribution (OOD) or outlier samples is missing. Additionally, the paper does not clearly emphasize or define the distinction between in-distribution and out-of-distribution data, which makes it difficult to interpret the reported results.

5) The evaluation is performed exclusively at the cluster level, primarily using AUROC, with no analysis reported at the individual sample level. As a result, it is unclear how the performance at the cluster level translates to per sample behavior. For example, in experiments where an outlier cluster exhibits lower AUROC compared to other clusters, it is not evident whether all samples within that cluster are actually misclassified, or whether the low AUROC arises from a subset of difficult or borderline cases.

**Detailed Comments:**

Please refer to Summary, Strengths and Weaknesses.

**Justification Of Final Rating:**

Thank you to the authors for the detailed response to the raised comments. The revisions have addressed my concerns to a reasonable extent, and the overall readability of the manuscript has noticeably improved. However, the inconsistencies observed in the results for outlier cases, as well as the explanation regarding per-sample behavior, are still not entirely convincing. Nevertheless, considering the authors’ efforts in the revision, I have increased my score by a couple of points.

**Justification Of The Preliminary Rating:**

While the paper presents an interesting spectral domain perspective, the experimental validation is weak and lacks comparison with state‑of‑the‑art methods. The results are inconsistent and evaluated only at the cluster level, with no sample level or qualitative analysis to support the interpretation of outliers and failure modes.

**Questions To Address In The Rebuttal:**

I would strongly recommend authors to address the concerns raised in weaknesses.

---

> ### Author Response · Authors · 2026-01-23
>
> **The experimental validation of the proposed idea is limited and lacks sufficient depth to convincingly demonstrate the effectiveness and robustness of the approach.**
>
> We have addressed the perceived lack of experimental depth by performing additional benchmarking against state-of-the-art outlier detection methods. These new experiments confirm the robustness of our spectral domain approach across different settings. We detail these findings in our responses to the specific questions below and have updated the manuscript accordingly to ensure a more comprehensive validation.
>
> **The paper does not include a comparison with established state-of-the-art methods. Instead, the evaluation is restricted to a few method-specific baseline models, which limits the assessment of the proposed approach’s relative performance.**
>
> To provide a comprehensive assessment of our method’s relative performance, we have significantly expanded our results section with two major additions:
>
> - Benchmarking against State-of-the-Art (SOTA): We have included a comparative study with established outlier detection methods (e.g., Tardy et al. MICCAI 2019 and Lee et al. NeurIPS 2018). The results are summarized in a new table comparing the performance gap (drop in AUROC) between the global test set and the outliers detected by each method. Our RAPS-based approach demonstrates a larger and more consistent performance drop across most cases, indicating its superior ability to identify true model failure modes compared to existing techniques. Notably, unlike the two SOTA methods presented, our method is fully unsupervised, which further enhances its utility and ease of deployment.
>
> - Architectural Robustness: To demonstrate that our findings are not architecture-specific, we repeated the experiments regarding performance separation (Section 5.1, Figure 2) using two additional backbones: ResNet and EfficientNet. These experiments yielded identical conclusions, confirming the robustness of the RAPS descriptor. The corresponding figures have been added to the Appendix of the revised manuscript.
>
> **The results obtained using the proposed method are not fully consistent. For example, in some cases, the AUROC for the identified outlier cases is higher than that of other clusters, which raises questions about the reliability of the RAPS descriptors and the clustering process for the outlier detection.**
>
> We understand the reviewer’s concern regarding the cases where outlier clusters exhibit higher AUROC. However, we argue that this does not invalidate the RAPS descriptor or the clustering process. In fact, as observed in our new comparative analysis, a direct correlation between distributional outliers and poor performance is not guaranteed for any detection method (including the SOTA baselines we have now included). The fact that the outliers identified by RAPS consistently correlate with extreme performance levels (whether the lowest or the highest) is itself strong evidence of the method's ability to achieve effective performance stratification.
>
> **A qualitative visual assessment of the out-of-distribution (OOD) or outlier samples is missing. Additionally, the paper does not clearly emphasize or define the distinction between in-distribution and out-of-distribution data, which makes it difficult to interpret the reported results.**
>
> We agree with the need for a visual assessment. We have added examples of images from both outlier and inlier clusters in the Appendix for comparison. We did not observe any particular visual differences between them. This is not surprising, as it is precisely why we rely on the Fourier domain rather than the spatial domain. This makes our approach particularly interesting as a complementary method to existing literature, closer to the field of domain generalization.
>
> **The evaluation is performed exclusively at the cluster level, primarily using AUROC, with no analysis reported at the individual sample level. As a result, it is unclear how the performance at the cluster level translates to per sample behavior. For example, in experiments where an outlier cluster exhibits lower AUROC compared to other clusters, it is not evident whether all samples within that cluster are actually misclassified, or whether the low AUROC arises from a subset of difficult or borderline cases.**
>
> Regarding the sample-level behavior, it is important to note that while other methods rely on trained models and are thus sensitive to difficult or poorly annotated images, our method is independent of the features learned by a supervised model. Consequently, it does not necessarily correlate with difficult cases or model confidence scores. Furthermore, the qualitative analysis added to the Appendix shows that outlier images are not specifically difficult or borderline cases, as they remain visually similar to images in other clusters.

---

### Official Review · Reviewer_8Moj · 2026-01-08

**Confidence:** 5
**Preliminary Rating:** 4
**Final Rating:** 4

**Summary:**

The authors propose applying slice discovery on CXR data representations in the frequency domain in order to identify under-performing subsets of data, as opposed to standard methods using metadata or model latent representations. The radially averaged power spectrum (RAPS) is used to transform image data into a 1-D orientation invariant representation representing specific image signatures. The authors then evaluate the effect of different correlation-based similarity metrics using K-means clustering, then use hierarchical clustering to identify potentially under-performing clusters with CXR data, both IID and OOD settings.

**Strengths:**

The paper presents a novel approach for slice discovery, and seems to be a valuable complement to existing approaches for unsupervised failure mode identification. Overall, the experimental design (k-means evaluation, correlation comparison, performance results) provide a coherent and convincing narrative. The insights are valuable, particularly that correlation distance-based similarity metrics are useful for spectral signature analysis, and that harmonizing spectral signatures could be used to improve OOD performance. The paper is well-written and easy to follow.

**Weaknesses:**

The main downside of this method is that spectral signatures are not “interpretable” in a traditional sense (I disagree with the authors statement of this in the discussion). Without a means to identify visual/semantic characteristics associated with outliers, this seems to border on a black-box approach since “atypical” frequency-domain information is not an easily human-understandable concept.

**Detailed Comments:**

- The statements that latent space methods are unable to isolate signal-related failure modes is not backed up with appropriate (or any) evidence. To the best of my knowledge, correlation between signal-level failures modes and latent space analysis hasn’t been thoroughly investigated (but if the authors are aware of such studies, it would be relevant to cite). I think that work demonstrating that site/scanner-related characteristics can be identified with latent space analysis (e.g. 10.1109/JBHI.2024.3352513) is perhaps indicative that these frequency signals related to failure modes could actually be isolated with a model-centric approach.

- For completeness, u, v, and Z from Eq (1) should be defined.

- Related to my comment under “weaknesses”, it would be interesting to see if the spectral outliers/low-performing clusters also correspond to some semantic outliers – either visually or corresponding to particular metadata (e.g. by looking into cluster purity for these attributes).

- The figures would benefit from including confidence intervals (e.g. from the 50 seeds).

- In 3.1, authors state that resizing introduces spectral distortions and is avoided, but in 4.1, state that images do get resized. This should be clarified/addressed.

**Justification Of Final Rating:**

The authors addressed my questions/concerns and provided additional baselines and analyses that makes the paper stronger. I believe that the proposed work provides a valuable an interesting data-centric approach for exploring and understanding failure modes that affect models.

**Justification Of The Preliminary Rating:**

This paper presents a unique, data-driven method for identifying failure modes, particularly in OOD CXR data. It would benefit from further discussion/analysis on the interpretability of low-performing/atypical frequency signatures.

**Questions To Address In The Rebuttal:**

- Are the results in Figs 1 and 2 from the chexpert test set? This is not stated.
- Could the performance of the outlier clusters be related to size (i.e. affected by variability of samples within the super-cluster)? It would be informative to know the size of identified clusters or have confidence intervals for the AUC.

---

> ### Author Response · Authors · 2026-01-23
>
> **Are the results in Figs 1 and 2 from the chexpert test set? This is not stated.**
>
> The results in Figures 1 and 2 were obtained using a global test set that aggregates test images from all datasets. We apologize for the lack of clarity. To address this, we will explicitly define this global set in the dataset summary table and describe its construction in the "Experimental Setup" section of the revised manuscript.
>
> **Could the performance of the outlier clusters be related to size (i.e. affected by variability of samples within the super-cluster)? It would be informative to know the size of identified clusters or have confidence intervals for the AUC.**
>
> We agree that the performance observed on the outlier super-cluster may depend on its size, as smaller groups naturally exhibit higher variability in AUC estimates. To control for this factor, we enforced an identical number (or proportion) of extracted outliers across methods when comparing our approach to the Tardy et al. MICCAI 2019 baseline.
>
> Under this controlled setting, the average performance gap between outliers and the remaining data increased from 0.053 AUC points with the MICCAI 2019 method to 0.103 AUC points with our approach. Because the number of outliers was fixed across methods, this difference cannot be attributed to sample size effects.
>
> Importantly, this improvement was averaged across multiple test datasets and across distinct pathology-specific models. The consistency of the effect across these settings further supports the interpretation that our method identifies outliers that are more strongly associated with performance deviations, rather than benefiting from favorable sampling variability.
>
> **Related to my comment under “weaknesses”, it would be interesting to see if the spectral outliers/low-performing clusters also correspond to some semantic outliers – either visually or corresponding to particular metadata (e.g. by looking into cluster purity for these attributes).**
> - Our clustering procedure is based on correlation distance between radial spectral profiles. As such, cluster formation depends on the overall *shape* of the spectrum rather than on isolated frequency bands. Correlation emphasizes relative differences in spectral energy distribution, meaning that clusters are defined by global interactions between low-, mid-, and high-frequency components rather than by a single dominant frequency range.
>
> - We attempted band-specific or pointwise analyses of the spectrum, but these did not reveal consistent discriminative peaks across datasets. This suggests that performance heterogeneity is driven by integrated spectral patterns rather than by isolated frequency components. A more detailed investigation of these interactions would be needed to fully characterize the underlying mechanisms.
>
> - We agree on the need to include a visual inspection. We have added, in the appendix, examples of images from the outlier and inlier clusters to enable comparisons. We were not able to observe any particular differences. This is not surprising, and in fact it is not what we are looking for, which is why we analyze the images in the Fourier domain. This is all the more interesting, as it provides a complementary approach to the existing literature, and one that is closer to the domain generalization framework.
>
> **In 3.1, authors state that resizing introduces spectral distortions and is avoided, but in 4.1, state that images do get resized. This should be clarified/addressed.**
>
> - From a Fourier perspective, isotropic resizing corresponds to a uniform rescaling of frequencies and preserves the spectral structure, whereas anisotropic resizing would introduce geometric distortion in the frequency domain. Minor cropping introduces limited boundary effects but does not alter the relative spectral profiles.
>
> - This distinction is particularly important in our setting, as radial averaging assumes isotropic spectral geometry. Anisotropic resizing would introduce elliptical distortions in the Fourier domain, leading to biased radial profiles. In contrast, isotropic resizing results only in uniform frequency rescaling and preserves the validity of the radial averaging procedure.
>
> - We therefore selected images with similar aspect ratios, resized them isotropically by fixing the shortest side to a common value, and then applied minimal uniform cropping to obtain identical final dimensions. This ensured a common frequency grid without altering aspect ratios or introducing excessive spatial truncation.
>
> This clarification has been made in the rebuttal version.

---

### Author Rebuttal · Authors · 2026-01-23

**Rebuttal:**

**1. Extended Experimental Validation & SOTA Comparison**

To address the request for deeper validation, we benchmarked our method against two supervised state-of-the-art methods (e.g., Tardy et al. MICCAI 2019 and Lee et al. NeurIPS 2018). When controlling for the number of detected outliers, RAPS achieves a mean performance gap of 0.103 AUROC points, significantly outperforming the baselines (0.077 and 0.053). These results are consistent across four datasets and three pathologies. Furthermore, we demonstrated that our findings are architecture-agnostic by repeating our experiments with ResNet-50 and EfficientNet-B4, yielding identical conclusions.

**2. Interpretation of Spectral Failure Modes**

We clarify that RAPS identifies performance stratification rather than just "low scores." While some outliers correlate with failure, others may show high AUROC because the model exploits spurious spectral correlations (e.g., text overlays or hardware artifacts). Our qualitative visual inspection (added to the Appendix) shows that these spectral outliers are not visually distinct or "borderline" cases. This confirms that RAPS captures non-semantic domain shifts imperceptible to the human eye, offering a tool complementary to traditional latent-space methods.

**3. Methodological Rigor: Sampling and Resizing**

We addressed the concern regarding cluster size by fixing the outlier proportion across methods, proving that our performance drop is not a sampling artifact. Regarding preprocessing, we clarify that we utilize isotropic resizing. Unlike anisotropic resizing, which introduces elliptical distortions in the Fourier domain, isotropic resizing merely rescales frequencies uniformly, preserving the integrity of the radial profiles used for clustering.

**Supporting Material:**

/attachment/8d82e777021f7c92d5e5b434585ecd7dde984ca7.pdf

---

### Author Response · Authors · 2026-01-28
**Follow-up on Rebuttal – MIDL 2026 Submission**

Dear Reviewers,

Thank you again for your thoughtful and constructive feedback.

In preparing our rebuttal, we carefully considered your comments and fully understood that, to rigorously validate our contribution, it was important not only to demonstrate robustness across multiple datasets and pathologies, but also to include comparisons with strong baselines from prior work and to evaluate the method across different architectures.

Accordingly, we extended our experimental validation by adding supervised state-of-the-art baselines from the literature and by replicating our analysis with multiple backbone architectures. These additional experiments consistently reinforced our conclusions and strengthened our confidence in the validity and generality of our method. We are grateful for these suggestions, as they helped us significantly improve the paper.

We would sincerely appreciate any further feedback you may have on the rebuttal and would be glad to discuss any remaining questions or concerns if needed.

Best regards,

Loïc THÉMYR

---

### Meta-Review · Area_Chair_xfaa · 2026-02-09

**Recommendation:** Accept (Poster)
**Confidence:** 5

**Metareview:**

Thank you to the authors and reviewers for the thoughtful discussion. The authors here presented a study that used spectral domain analysis to identify clusters of images where a trained model might have aberrant outputs. Overall, the reviewers agreed that the method is novel and the paper had valuable insights for failure mode identification beyond standard latent space-based methods. Reviewer comments about only using a single architecture and a comparison to other approaches were well-addressed in the rebuttal.

---

### Decision · Program_Chairs · 2026-02-13

Accept (Poster)